# Receptor-Specific Diffusion Model: Towards Generating Protein-Protein Structures with Customized Perturbing and Sampling

## Abstract

Recent advancements in deep generative models have significantly facilitated protein-ligand structure design, which is crucial in protein engineering. However, recent generative approaches based on diffusion models in this field usually start sampling from a unified distribution, failing to capture the intricate biochemical differences between receptors. This may limits their capacity to generate reliable ligands for the corresponding receptors. Moreover, the current sampling process incurs a heavy computational burden and inefficiency, which further escalates the training demands on the model. To this end, we introduce a novel diffusion model with customized perturbing and sampling for the protein-ligand design targeting the specific receptor, named as Receptor-Specific Diffusion Model (RSDM). In particular, the receptor-specific information is used to tailor fine-grained sampling distributions via changing the noise for customized perturbing. Meantime, we refine the sampling process using a predefined schedule to perform stepwise denoising and gradually decrease the influence of the receptor's guidance in the ligand generation for customized sampling. The experimental reaults indicate that RSDM is highly competitive with state-of-the-art learning-based models, including the latest models like ElliDock and DiffDock-PP. Additionally, RSDM stands out for its faster inference speed compared with all baseline methods, highlighting its potential for generating dependable protein-ligand.

## 1 Introduction

Protein design is essential in biomedical research, particularly for targeting specific proteins, by facilitating the development of highly specific drugs and deepening our understanding of biological mechanisms. Protein-ligand structure design complements protein design by providing insights into how a designed protein will interact with its receptor, such as drugs or substrates. To accurately predict protein-ligand structures for a given protein-receptor, researchers need to determine ligand-bound conformations that are specific to the receptor while ensuring the stability and functionality of the resulting complex. Traditional search-based methods (Chen et al., 2003; De Vries et al., 2010; de Vries et al., 2015) employ a scoring function paired with search techniques to identify the most plausible predicted pose of a ligand matching experimental data. While these methods can yield satisfactory results, they are computationally intensive and time-consuming.

Recently powerful learning-based methods (Gainza et al., 2019; Ganea et al., 2021; Yu et al., 2024; Ketata et al., 2023; Guan et al., 2024; Evans et al., 2021) aim to predict the final pose of the input ligand directly, prioritizing an end-to-end, data-driven approach. Deep learning generation methods based on diffusion are gaining increasing attention due to their global 3D structure generation capability and ability to rapidly produce multiple conformations simultaneously. These methods formulate protein-ligand structure design as a generative problem: given an interacting protein pair, the goal is to estimate the distribution over all potential poses using a diffusion model. For example, DiffDock-PP (Ketata et al., 2023) aims to directly predict the structure of the protein-ligand while comprehensively considering both the ligand pose and the protein-receptor structure.

However, we have identified issues in the current diffusion process that may limit its performance for ligand structure design. In the forward process, applying noise sampled from a unified distri-

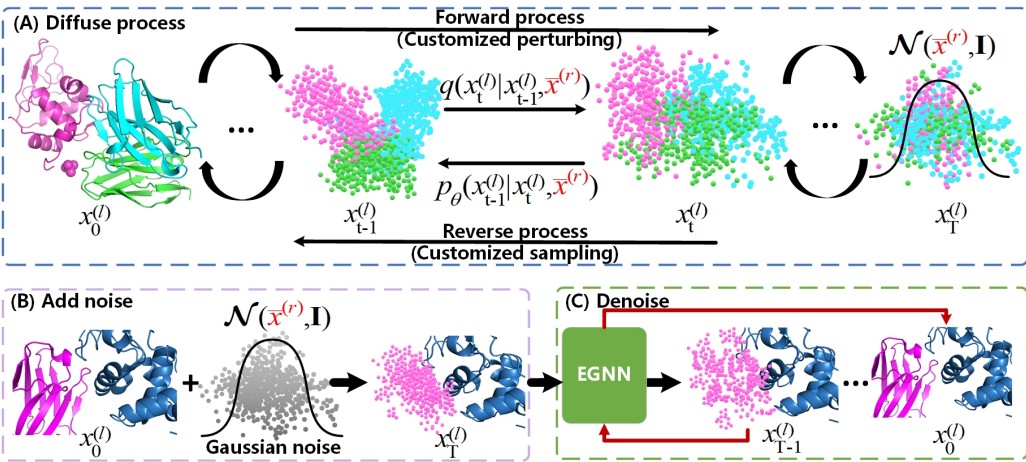

Figure 1: Diagram of RSDM. (A) The overall workflow of the receptor-specific diffusion process, refined through customized perturbing and sampling. (B) The forward process of the receptor-specific diffusion process, where random noise is sampled from a personalized sampling distribution $\mathcal{N}(\bar{\mathbf{x}}^{(r)}, \mathbf{I})$, based on the corresponding receptor, and added to $\mathbf{x}_0^{(l)}$ to obtain $\mathbf{x}_T^{(l)}$. (C) The reverse process of the receptor-specific diffusion process, where EGNN gradually recovers a realistic structure $\mathbf{x}_0^{(l)}$ from initial random noise $\mathbf{x}_T^{(l)}$ conditioned on the receptor.

bution to each ligand fails to identify the inherent differences between receptors, overlooking their unique structural and chemical properties. In the reverse process, most canonical diffusion-based models require predicting the noise-free data from its current noisy version and then estimating its noisy version at the previous time step. This two-step estimation process complicates the training process and fails to account for the specific receptor's guiding role in ligand generation, neglecting its influence on producing accurate ligand structures.

To optimize the diffusion process mentioned above, as shown in Figure 1, we propose a novel receptor-specific diffusion model (RSDM) towards generating ligand structures with customized perturbing in the forward process and customized sampling in the reverse process. Specifically, RSDM refines the diffusion process using two targeted strategies to enhance both the accuracy and computational efficiency of diffusion-based approaches. In the forward process, *personalized sampling distribution* applies customized noise perturbation for each ligand by tailoring the noise according to receptor-specific information. In the reverse process, the RSDM employs customized sampling via *step-by-step data purification* to iteratively refine the model's output based on a predefined schedule that incorporates receptor-specific information. This schedule enables the model to directly predict the noise-perturbed sample from the previous time step based on the current sample while gradually reducing the receptor's influence on ligand generation. This refined diffusion process offers two key benefits: First, customized perturbing ensures that the generated ligand is strongly influenced by its corresponding receptor during the initial phases of the sampling process, which is crucial for maintaining receptor-ligand specificity. Second, customized sampling prevents over-reliance on receptor guidance, allowing the model to generate a independent ligand structure that is more biologically accurate and functional. Our experimental results demonstrate that RSDM exhibits robust competitiveness against state-of-the-art learning-based models, while significantly reducing inference times compared to all baseline methods.

## 2 RELATED WORK

### 2.1 PROTEIN-PROTEIN DOCKING

The existing complex structures capture merely a fraction of the vast number of interactions believed to occur within living organisms. Manually collecting and labeling a sufficient amount of protein

complexes data is impractical due to its time-consuming and labor-intensive property. Thus it is highly necessary to discover effective and novel protein complexes to development protein-protein docking experimental efforts with computational approaches. Traditional docking methods (Chen et al., 2003; De Vries et al., 2010; de Vries et al., 2015; Yan et al., 2020) follow the scheme that typically begins by sampling from the geometric space of the two interacting proteins, then use a scoring function to assess binding affinity, and finally refine the structures obtained in earlier stages using an energy model. Recently deep learning methods for protein-protein docking task can be roughly classified into two groups, i.e., single-step and multi-step methods. The former (Ganea et al., 2021; Sverrisson et al., 2022; Watson et al., 2023) predicts the complex structure directly in one step, while the latter (Evans et al., 2021; McPartlon & Xu, 2023; Guan et al., 2024) iteratively refines a set of proposed structures to produce its final predictions.

## 2.2 Equivariant Graph Neural Networks (EGNNs).

Due to any problems exhibit 3D translation and rotation symmetries, such as point clouds (Uy et al., 2019) and 3D molecular structures (Ramakrishnan et al., 2014), it is often desired that predictions on these tasks are either equivariant or invariant with respect to different coordinate transformation. Recent works (Fuchs et al., 2020; Jiao et al., 2023; Jing et al., 2021; Satorras et al., 2021; Yim et al., 2023) are proposed from geometric first-principles to improve the ability of traditional GNNs on achieving equivariance from E(3) transformations. SE(3)-Transformers (Fuchs et al., 2020) employs the equivariance constraints on the self-attention to ensure the output of model is invariant to global rotations and translations. EGNN (Satorras et al., 2021) computes the weight coefficient via the relative squared distance between particles to guarantee equivariance, without requiring the spherical harmonics (Fuchs et al., 2020; Thomas et al., 2018). FrameDiff (Yim et al., 2023) implements the proposed theory as a SE(3) invariant diffusion model for protein backbone generation.

## 2.3 Diffusion Models

Diffusion models (Sohl-Dickstein et al., 2015; Ho et al., 2020; Song & Ermon, 2019) are increasingly powerful tools to generate novel and effective samples by iteratively denoising data points sampled from a prior noise distribution, which have shown unprecedented success in images (Dhariwal & Nichol, 2021; Nichol et al., 2021) and texts (Ramesh et al., 2022). Considering the great potential of diffusion models in generating data, several recent works (Dhariwal & Nichol, 2021; Ho & Salimans, 2022) have proposed expanding diffusion models to generate protein structures. ProteinSGM (Lee et al., 2023) implements the diffusion process via learning inter-residue 6D coordinates in an amino acid chain based on the idea of the score-based diffusion model (Song et al., 2020). FoldingDiff (Wu et al., 2024) implements the diffusion model on the inter-residue angles in protein backbones instead of 3D coordinates. Due to the primary objective of the initial diffusion model is to understand the data distribution, some researchers incorporate classifier-based guidance to implement controllable generation. DiffSBDD (Schneuing et al., 2022) employs the diffusion model to design small-molecule ligands while keeping SE(3)-equivariance. DiffAb (Luo et al., 2022) develops a deep learning model to generate antibodys explicitly by considering the 3D information from antigens.

## 3 Preliminaries

Denoising diffusion probabilistic models (DDPM) involves analyzing a real data distribution $q(\mathbf{x})$ and a sample $\mathbf{x}_0$ taken from it. During the forward process, Gaussian noise is incrementally introduced to the sample over $T$ steps, which is akin to a Markov chain. This process generates a sequence of noisy samples $\mathbf{x}_1, \cdots, \mathbf{x}_T$, with the subscript $t$ representing the diffusion timestep and a pre-defined variance schedule $\beta_1, \cdots, \beta_T$:

$$q(\mathbf{x}_t|\mathbf{x}_{t-1}) := \mathcal{N}(\mathbf{x}_t; \sqrt{1-\beta_t}\mathbf{x}_{t-1}, \beta_t\mathbf{I}), q(\mathbf{x}_{1:T}|\mathbf{x}_0) := \prod_{t=1}^{T} q(\mathbf{x}_t|\mathbf{x}_{t-1}). \qquad (1)$$

The reverse process attempts to invert the forward process by learning a parameterized model on a conditional distribution $p_\theta(\mathbf{x}_{t-1}|\mathbf{x}_t)$. It is also a Markov chain, but it runs in the opposite direction,

from the noise distribution back to the original data distribution:

$$p_\theta(\mathbf{x}_{t-1}|\mathbf{x}_t) := \mathcal{N}(\mathbf{x}_{t-1}; \mu_\theta(\mathbf{x}_t, t), \Sigma_\theta(\mathbf{x}_t, t)), p_\theta(\mathbf{x}_{0:T}) := p(\mathbf{x}_T) \prod_{t=1}^{T} p_\theta(\mathbf{x}_{t-1}|\mathbf{x}_t), \quad (2)$$

where $p(\mathbf{x}_T) \sim \mathcal{N}(\mathbf{0}, \mathbf{I})$. The parameter $\theta$ is optimized by maximizing the evidence lower bound, defined as $\mathbb{E}_q \left[ \ln \frac{p_\theta(\mathbf{x}_{0:T})}{q(\mathbf{x}_{1:T}|\mathbf{x}_0)} \right]$ (Jordan et al., 1999; Blei et al., 2017). Sampling from the diffusion model involves first drawing a sample from $p(\mathbf{x}_T)$ and then running the reverse diffusion process, transitioning step-by-step from $t = T$ to $t = 0$. Additionally, diffusion models can be easily extended to conditional models by conditioning the reverse process on some context $c$, resulting in $p_\theta(\mathbf{x}_{t-1}|\mathbf{x}_t, c)$.

## 4 METHODS

### 4.1 DEFINITIONS AND NOTATIONS

In this work, our proposed model aims to generate a protein-ligand that can bind to a given protein-receptor. Both the generated ligand and the receptor are modeled at the residue level. We define a graph denoted as $\mathcal{G} = (\mathcal{V}, \mathcal{E})$ to represent a protein. Each node $v_i \in \mathcal{V}$ denotes the $i$-th residue with a tuple $(\mathbf{h}_i, \mathbf{x}_i)$, where $\mathbf{h}_i \in \mathbb{R}^d$ denotes the SE(3)-invariant embedding and $\mathbf{x}_i \in \mathbb{R}^{14 \times 3}$ is the 3D coordinate of all atoms in the $i$-th residue. The collection of all nodes yields $\mathbf{H}^{(l)} \in \mathbb{R}^{n \times d}$ and $\mathbf{X}^{(l)} \in \mathbb{R}^{n \times 14 \times 3}$ for representing the ligand, composed of $n$ residues. Similarity, $\mathbf{H}^{(r)} \in \mathbb{R}^{m \times d}$ and $\mathbf{X}^{(r)} \in \mathbb{R}^{m \times 14 \times 3}$ are used to represent the receptor, composed of $m$ residues. We fix the receptor $\mathbf{X}^{(r)}$ and leverage it to predict the structure of the ligand with respect to this receptor. In this way, the task of generating ligand structures can be formulated as a 3D point cloud completion task. The ground-truth $\mathbf{X}^{(l)^*}$ is leveraged to evaluate the docking performance via comparing it with $\widetilde{\mathbf{X}}^{(l)}$, where $\widetilde{\mathbf{X}}^{(l)}$ denotes the model's prediction.

### 4.2 PROTEIN DIFFUSION MODEL IN 3D

Our proposed model is based on DDPM, which employs a Markov process to introduce random noise to a sample $\mathbf{x}_0$ across $T$ discrete time steps until it becomes indistinguishable denoted as $\mathbf{x}_T$. Recent advancements in modeling 3D data have demonstrated that neural networks built to follow geometric invariances can introduce meaningful biases, thereby enhancing model generalizability and training efficiency (Batzner et al., 2022). Motivated by this insight, we incorporate an equivariant graph neural network (EGNN) (Satorras et al., 2021) into the diffusion model as $f_\theta$, which demonstrates equivariance to transformations within the Euclidean group when handling 3D data.

Before generating ligand structures, we need to encode the input point cloud with atoms to capture the underlying structural dependencies between the ligand and the receptor. Specifically, we construct a two-level encoder (Jin et al., 2022) to capture ligand-receptor interactions, including an atom-level encoder and a residue-level encoder.

- The atom-level encoder takes atom types as model input and constructs a $K$ nearest neighbor graph for each atom. The edge embeddings between two atoms are derived from two perspectives: radial basis function and position embedding. $e_{uv}^{(0)} = \text{RBF}(||\mathbf{x}_u - \mathbf{x}_v||)$ denotes the edge embedding derived according to the radial basis computed based on the distance between two atoms $u$ and $v$. While $e_{uv}^{(1)} = Pe(pos_u, pos_v)$ represents the edge embedding learned from the position embedding (Vaswani et al., 2017). Subsequently, the final edge embedding $e_{uv}$ can be obtained by: $e_{uv} = e_{uv}^{(0)} \oplus e_{uv}^{(1)}$, where $\oplus$ signifies the concatenation operation.

- The residue-level encoder constructs a $K$ nearest neighbor graph for each residue. After pooling all atom embeddings belonging to the same amino acid and concatenating the resulting embedding with the dihedral angle embedding obtained by calculating the angles between the backbone atoms (N, $C_\alpha$, C) with cosine function, a residue-level structure embedding is derived. Additionally, the residue-encoder learns the semantic embedding of each residue based on chemical properties such as polarity, hydropathy and so on. Considering the edge embedding between a pair of residues,

the key distinction from the atom-level encoder lies in the incorporation of orientation feature $O_i \in \mathrm{SO}(3)$.

The outputs of the two-level encoder are leveraged in the message passing process of EGNN to update SE(3)-invariant embeddings $\mathbf{h}$ and predict 3D atom coordinates $\mathbf{x}$. To enable EGNN to predict the ligand structure given the corresponding receptor, we identify the $Z$ nearest neighbor residues of the receptor to determine the binding sites $\mathcal{P} = (\mathbf{h}_i, \mathbf{x}_i)_{i \in 1, \ldots, Z}$. The number of binding sites $Z$ is a hyper-parameter. Subsequently, EGNN is employed to encode and predict the structure of the ligand based on the binding sites, thereby realizing the conditional prediction.

## 4.3 RECEPTOR-SPECIFIC DIFFUSION POLICY

Most canonical diffusion-based models for protein design aim to reconstruct corrupted (noised) protein structures and generate new ones by reversing the corruption process. This is achieved through iterative denoising $\mathbf{x}_T$, transforming initial random noise $\mathbf{x}_T$ into a realistic protein $\mathbf{x}_0$. Our receptor-specific diffusion model (RSDM) employs a tailored diffusion policy that adapts both the forward and reverse processes for more accuracy and efficient ligand structure generation.

**Personalized sampling distribution (PSD).**

The use of noise sampled from a unified distribution, without accounting for receptor differences, poses a significant challenge for receptor-specific ligand generation. To address this, we propose modifying the sampling process by introducing a receptor-specific personalized noise distribution. The motivation behind this refinement is to ensure that the receptor plays a dominant role in shaping the noise at the initial timestep of sampling, thereby maintaining receptor-specificity. The experimental results obtained using RMSD (Root Mean Square Deviation) loss on $\mathbf{C}_\alpha$, as shown in Figure 2, validate our above-mentioned motivation. Specifically, we adjust the traditional diffusion model's sampling from $\mathbf{x}_T \sim \mathcal{N}(\mathbf{0}, \mathbf{I})$ to $\mathbf{x}_T \sim \mathcal{N}(\bar{\mathbf{x}}^{(r)}, \mathbf{I})$

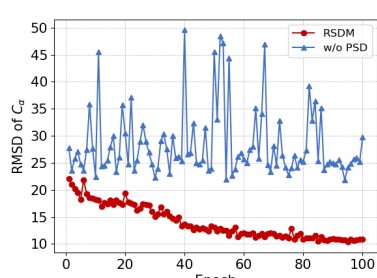

Figure 2: RMSD loss curves of $\mathbf{C}_\alpha$ for different methods on SAbDab dataset.

to create a personalized sampling distribution for each ligand, where $\bar{\mathbf{x}}^{(r)} \in \mathbb{R}^{1 \times 3}$ denotes the mean value of the 3D atomic coordinates of the corresponding receptor associated with the binding sites $\mathcal{P}$. To implement this, in the forward process, we incorporate the receptor-specific information as additional context in the forward process, extending the forward diffusion process described in Eq. 1 as follows:

$$q(\mathbf{x}_t^{(l)} | \mathbf{x}_{t-1}^{(l)}, \bar{\mathbf{x}}^{(r)}) := \mathcal{N}(\mathbf{x}_t^{(l)}; \sqrt{1-\beta_t}\mathbf{x}_{t-1}^{(l)} + \frac{\gamma_t}{T}\bar{\mathbf{x}}^{(r)}, \beta_t\mathbf{I}), \quad (3)$$

where $\beta_t$ denotes a pre-defined variance schedule and $\gamma_t$ represents the impact coefficient at the timestep $t$. Since we aim to adjust the original sampling distribution of the diffusion model from $\mathcal{N}(\mathbf{0}, \mathbf{I})$ to $\mathcal{N}(\bar{\mathbf{x}}^{(r)}, \mathbf{I})$, $\mathbf{x}_t^{(l)} = \sqrt{\bar{\alpha}_t}\mathbf{x}_0^{(l)} + \sqrt{1-\bar{\alpha}_t}\epsilon$ is extended as:

$$\mathbf{x}_t^{(l)} = \sqrt{\bar{\alpha}_t}\mathbf{x}_0^{(l)} + \sqrt{1-\bar{\alpha}_t}\epsilon + \sum_{i=1}^{t} (\prod_{j=i+1}^{t} \sqrt{\alpha_j})\frac{\gamma_i}{T}\bar{\mathbf{x}}^{(r)}, \quad (4)$$

where $\alpha_j = 1 - \beta_j$ and $\bar{\alpha}_t := \prod_{s=1}^{t} \alpha_s$. Refer to Appendix A for a detailed derivation of Eq. 4. The schedule of traditional diffusion models is updated to incorporate $\gamma$ in the formulation of $\mathbf{x}_t^{(l)}$. This update serves the purpose of integrating receptor-specific information into the diffusion process, providing better control over its impact on the generated outputs. The schedule of $\gamma_t$ is defined as:

$$\gamma_t = \frac{1}{\prod_{j=t+1}^{T} \sqrt{\alpha_j}} = \frac{1}{\sqrt{\frac{\bar{\alpha}_T}{\bar{\alpha}_t}}}. \quad (5)$$

When $t = T$, we have $\bar{\alpha}_T := 0$, $\prod_{j=T+1}^{T} \sqrt{\alpha_j} = 1$ and $\sum_{i=1}^{T}(\prod_{j=i+1}^{T} \sqrt{\alpha_j})\frac{\gamma_i}{T}\bar{\mathbf{x}}^{(r)} = \bar{\mathbf{x}}^{(r)}$. Therefore, $\mathbf{x}_t^{(l)} \sim \mathcal{N}(\bar{\mathbf{x}}^{(r)}, \mathbf{I})$ since $\epsilon \sim \mathcal{N}(\mathbf{0}, \mathbf{I})$.

**Step-by-step data purification (SDP).** After the forward process, we next discuss the reverse process, which goes from $t = T$ to 0. In most existing diffusion models designed for proteins (Trippe

et al., 2022; Watson et al., 2023), the reverse process often proceeds with the model predicting $\mathbf{x}_0$ from the input $\mathbf{x}_t$ and then deriving $\mathbf{x}_{t-1}$, which can be formulated as:

$$\mu_\theta(\mathbf{x}_t^{(l)}, t) = \frac{1}{\sqrt{\alpha_t}}(\mathbf{x}_t^{(l)} - \frac{\beta_t}{\sqrt{1 - \bar{\alpha}_t}}\widetilde{\epsilon}), \tag{6}$$

$$\widetilde{\epsilon} = (\mathbf{x}_t^{(l)} - \sqrt{\bar{\alpha}_t}f_\theta(\mathbf{x}_t^{(l)}, t))/\sqrt{1 - \bar{\alpha}_t}, \tag{7}$$

where $f_\theta$ denotes the EGNN introduced in Subsection 4.2. Such the reverse process poses a challenge to the model's predictive ability and complicates the training process. Therefore, in our RSDM, the schedule of traditional diffusion models is updated not only to follow the progressive denoising process from $\mathbf{x}_T$ to $\mathbf{x}_0$, but also to systematically diminish the influence of receptor-specific information throughout the schedule of $\gamma$. Specifically, when given $\mathbf{x}_t^{(l)}$ and current time $t$, we utilize $f_\theta(\mathbf{x}_t^{(l)}, t)$ to directly predict $\mathbf{x}_{t-1}^{(l)}$ under the guidance of $\bar{\mathbf{x}}^{(r)}$: $\mathbf{x}_{t-1}^{(l)} = f_\theta(\mathbf{x}_t^{(l)}, t, \bar{\mathbf{x}}^{(r)}) \sim p_\theta(\mathbf{x}_{t-1}^{(l)}|\mathbf{x}_t^{(l)}, t, \bar{\mathbf{x}}^{(r)})$, where $p_\theta(\mathbf{x}_{t-1}|\mathbf{x}_t)$ from Eq. 2 can be extended as:

$$p_\theta(\mathbf{x}_{t-1}^{(l)}|\mathbf{x}_t^{(l)}, t, \bar{\mathbf{x}}^{(r)}) := \mathcal{N}(\mathbf{x}_{t-1}; \mu_\theta(\mathbf{x}_t^{(l)}, t) - \frac{\gamma_t}{T}\bar{\mathbf{x}}^{(r)}, \Sigma_\theta(\mathbf{x}_t^{(l)}, t)), \tag{8}$$

This gradual and sequential denoising process iteratively refines the denoising results, reducing the reliance on the model's strong predictive capabilities for producing satisfactory results. Meantime, this refined reverse process enables the model to gradually shift its focus away from the receptor and towards refining the ligand structure independently. It's important to note that such step-by-step data purification may increase computational overhead compared to typical generative diffusion—requiring computation $T$ times in a training epoch. However, incorporating receptor-specific information can effectively guide ligand generation, allowing for fewer diffusion steps. The results shown in Tables 1 and 2 demonstrate that our model achieves satisfactory performance even with a single-digit value for $T$. This indicates that the refined diffusion process can significantly reduce the computational burden of the model during sampling, thereby improving its computational efficiency.

## 4.4 MODEL OPTIMIZATION

We design two loss functions, namely the reconstructed structure loss and the reconstructed coordinate loss, as the objective function for model parameter optimization.

**Reconstructed structure loss.** Reconstructed structure loss comprises five distinct types of loss designed to ensure the reliability of the generated ligands: (1) $\mathcal{L}_{local}$ calculates the spatial distances among all atoms within the same amino acid; (2) $\mathcal{L}_{global}$ computes the spatial distances among all atoms between a ligand and a receptor; (3) $\mathcal{L}_{local}^{C_\alpha}$ measures the distances between all $C_\alpha$ atoms across all amino acids in the ligand; (4) $\mathcal{L}_{global}^{C_\alpha}$ evaluates the distances between all $C_\alpha$ atoms between a ligand and a receptor; and (5) $\mathcal{L}_{angle}$ quantifies the disparity between the predicted and the gound-truth dihedral angles. The objective function used to compute $\mathcal{L}_{angle}$ is an expected MSE loss:

$$\mathcal{L}_{\text{MSE}}(\mathbf{V}, \widetilde{\mathbf{V}}) = \frac{1}{n}\sum_{i=1}^{n}(\mathbf{v}_i - \widetilde{\mathbf{v}}_i)^2, \tag{9}$$

where $\mathbf{V}$ denotes the ground-truth dihedral angles and $\widetilde{\mathbf{V}}$ denotes the predictions of the model. The vector $\mathbf{v}_i$ signifies the predicted dihedral angles for the $i$-th residue. Other above-mentioned four types of loss $\mathcal{L}_{local}$, $\mathcal{L}_{global}$, $\mathcal{L}_{local}^{C_\alpha}$, and $\mathcal{L}_{global}^{C_\alpha}$ are computed with Huber loss, which can be formulated as:

$$\mathcal{L}_{\text{HuberLoss}}(y, \widetilde{y}) = \begin{cases} \frac{1}{2}(y - \widetilde{y})^2 & if|y - \widetilde{y}| \leq \delta \\ \delta(|y - \widetilde{y}| - \frac{1}{2}\delta) & \text{otherwise}, \end{cases} \tag{10}$$

where $y$ and $\widetilde{y}$ represent the ground-truth and model predictions, respectively. $\delta$ is a hyper-parameter used to control the balance between the squared loss and the absolute loss.

**Reconstructed coordinate loss.** The objective of the reconstructed coordinate loss is to minimize the expected KL divergence between the distribution of Eq. 3 and Eq. 8:

$$\mathcal{L}_{\text{coordinate}} = \mathbb{E}_q\left[\sum_{t=1}^{T} D_{\text{KL}}(q(\mathbf{x}_{t-1}^{(l)}|\mathbf{x}_t^{(l)}, \mathbf{x}_0^{(l)}, \bar{\mathbf{x}}^{(r)})||p_\theta(\mathbf{x}_{t-1}^{(l)}|\mathbf{x}_t^{(l)}, \bar{\mathbf{x}}^{(r)}))\right] \tag{11}$$

The training process of RSDM is summarized as Algorithm 1 in Appendix B.

## 5 EXPERIMENTS

### 5.1 EXPERIMENTAL SETUP

▷ **Datasets**. We evaluate RSDM on two datasets:

**Docking benchmark version 5 (DB5.5).** DB5.5 (Vreven et al., 2015) is recognized as a gold standard dataset for its high-quality data, encompassing 253 high-quality complex structures. Following the data partitioning approach of EquiDock (Ganea et al., 2021), DB5.5 is divided into training, validation, and test sets with sizes of 203, 25, and 25, respectively.

**The Structural Antibody Database (SAbDab).** SAbDab (Dunbar et al., 2014) is a specialized database curated for ligand-receptor complexes. The data is partitioned based on sequence similarity assessed by MMseqs2 (Steinegger & Söding, 2017), resulting in a training set and a validation set with sizes of 1781 and 300, respectively. For performance evalution, we employ an independent test set with 54 ligand-receptor complexes curated from the RAbD (Adolf-Bryfogle et al., 2018) database. The setting of the evaluation tests on SAbDab in this work aligns with that of Ellidock (Yu et al., 2024).

▷ **Baselines**. To verify the effectiveness of RSDM, we compare it with five state-of-the-art methods for protein-protein docking, including the alphafold-based protein complex prediction model Alphafold-Multimer (Evans et al., 2021), the template-based docking server HDock (Yan et al., 2020), the regression-based docking model EquiDock (Ganea et al., 2021), the diffusion-based docking model DiffDock-PP (Ketata et al., 2023) and interface-fitting approach docking model Ellidock (Yu et al., 2024). The recommended hyperparameters of EquiDock, DiffDock-PP, and Ellidock are applied in our evaluation tests. The original pre-trained models are used for HDock and Alphafold-Multimer.

▷ **Implementation**. Our models are trained and tested on NVIDIA A40 GPUs, each with 48GB of memory. The hierarchical encoder consists of four message passing layers to update the target node embedding with a hidden dimension of 256. We utilize the Adam optimizer (Kingma & Ba, 2015) with a learning rate of $1 \times 10^{-3}$. The dropout ratio is set to 0.1. The number of nearest neighbors $K$ is set to 9. RSDM is trained with $\beta_1 = 1 \times 10^{-4}$, $\beta_T = 0.7$, and $T = 8$ for 500 epochs. We save the model with the lowest loss evaluated on the validation set. The ligand structure generated by RSDM is refined using OpenMM (Eastman et al., 2017) and then be utilized for performance evaluation.

▷ **Evaluation metrics**. To ensure a fair comparison, we follow the evaluation metrics used in Ellidock (Yu et al., 2024), containing Complex Root Mean Squared Deviation (CRMSD), Interface Root Mean Squared Deviation (IRMSD) and DockQ (Basu & Wallner, 2016). The details of these evaluation metrics are introduced in Appendix C.

### 5.2 COMPARISONS OF THE DOCKING PERFORMANCE

**Q: Whether RSDM can outperform the baseline methods that do not rely on searching mechanisms?** Yes, RSDM has shown promising results compared to the baseline methods that do not rely on searching mechanisms. The key advantage of RSDM lies in integrating receptor-specific information directly into the diffusion process, enabling it to capture complex interactions more effectively—a limitation present in many current diffusion processes.

We assess the docking performance of different methods on two datasets DB5.5 and SAbDab. Experimental results are shown in Tables 1 and 2 for each respective dataset. From Tables 1 and 2, we observe that ❶ RSDM outperforms all the baseline methods without searching, including EquiDock, DiffDock-PP, and Ellidock, across almost all evaluation metrics on both DB5.5 and SAbDab datasets. These results demonstrate our model's efficacy in tackling protein-protein docking challenges. By excelling across multiple evaluation metrics, our model ensures a holistic advantage, offering a dependable solution for addressing complex protein-protein docking tasks. ❷ It is notable that the mean scores of some models surpasses the corresponding median scores, whereas our model exhibits mean scores lower or closer in comparison with its median scores. This discrepancy suggests that while other models may excel in specific scenarios, our model showcases a more robust overall performance, less susceptible to the influence of extreme values. This observation indicates the superior adaptability of our model's adaptability across diverse docking scenarios,

Table 1: **Complex prediction results (DB5.5 test).** Note that * means we use the pre-trained model for testing, otherwise we train the model from scratch on the corresponding training set before testing. The best results for methods without searching are in bold, and the second-best results are underlined.

| Method / Metric | | With Searching | | Without Searching | | | |
|---|---|---|---|---|---|---|---|
| | | HDock* | Multimer* | EquiDock | DiffDock-PP | ElliDock | Ours |
| **CRMSD(↓)** | median | 0.327 | 1.987 | 14.136 | 14.109 | 12.995 | **10.044** |
| | mean | 3.745 | 7.081 | 14.726 | 15.419 | 14.413 | **10.626** |
| | std | 7.139 | 7.258 | 5.312 | 8.160 | 6.780 | 4.331 |
| **IRMSD(↓)** | median | 0.289 | 1.759 | 11.971 | 15.060 | 11.134 | **8.282** |
| | mean | 3.548 | 7.141 | 13.233 | 16.881 | 12.480 | **8.550** |
| | std | 6.842 | 7.889 | 4.931 | 11.397 | 4.966 | 1.955 |
| **DockQ(↑)** | median | 0.981 | 0.629 | 0.036 | 0.025 | 0.037 | **0.166** |
| | mean | 0.791 | 0.482 | 0.044 | 0.035 | 0.060 | **0.159** |
| | std | 0.386 | 0.418 | 0.034 | 0.033 | 0.060 | 0.049 |
| **Inference time** | | 11478.4 | 56762.5 | 60.1 | 2103.1 | 36.7 | **5.2** |

thus it can provide consistent and reliable outcomes. ❸ For the comparison with the search-based methods, although HDock yields the best results, there might be potential data leakage issues due to its predictive template-based modeling approach (Yu et al., 2024). Similarly, Multimer extends AlphaFold to support multiple chains, inheriting its powerful representation capabilities achieved through the integration of various methods, such as multiple sequence alignments (MSAs) of homologous sequences. Moreover, our method is significantly more efficient than these two methods. Further details on this efficiency can be found in Subsection 5.3.

To provide a more intuitive comparison, we visualize the distributions of CRMSD and IRMSD for each method in Figure 3. Additionally, to illustrate the superiority of RSDM in prediction accuracy, scatter plots of data distribution using DockQ as the evaluation metric on the SAbDab dataset are presented in Figure 4. Additional scatter plots on the DB5.5 dataset are presented in Figure 5. As depicted in Figures 3 and 4, we observe that ❶ RSDM exhibits a relatively symmetric distribution with a moderate spread, suggesting a balanced performance across different docking scenarios. In contrast, other models present narrower and taller distributions, implying higher consistency but potentially limited adaptability to diverse protein-protein interactions. ❷ RSDM displays a shorter tail, suggesting its more consistent docking performance. While other models exhibit relatively elongated tails, indicating that these method can fail to provide reasonable results in certain specific docking scenarios. ❸ The results in Figure 4 show that most data points are consistently clustered in the lower right quadrant of the dashed line, demonstrating a higher level of precision and reliability of RSDM in protein-protein docking compared to baseline methods.

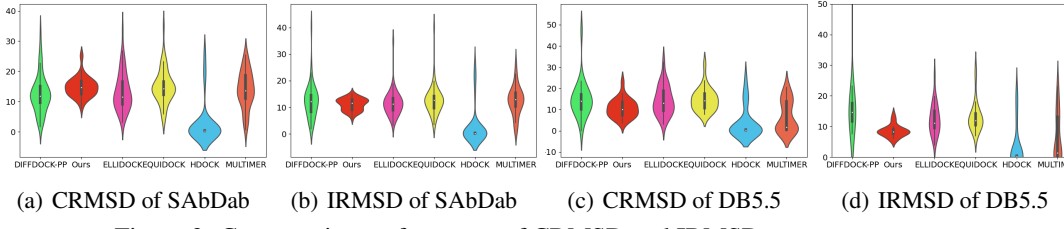

| (a) CRMSD of SAbDab | (b) IRMSD of SAbDab | (c) CRMSD of DB5.5 | (d) IRMSD of DB5.5 |

Figure 3: Comparative performance of CRMSD and IRMSD on two test sets.

## 5.3 COMPARISONS OF THE INFERENCE TIME

**Q: Whether RSDM's inference time is competitive with all baseline methods?** Yes, RSDM's inference time is superior to that of the baseline methods. The receptor-specific information enhances the guidance for ligand generation, allowing the model to converge more quickly and efficiently during inference.

Table 2: **Complex prediction results (SAbDab test).** Note that * means we use the pre-trained model for testing, otherwise we train the model from scratch on the corresponding training set before testing. The best results for methods without searching are in bold, and the second-best results are underlined.

| Method Metric | | With Searching | | Without Searching | | | |
|---|---|---|---|---|---|---|---|
| | | HDock* | Multimer* | EquiDock | DiffDock-PP | ElliDock | Ours |
| **CRMSD($\downarrow$)** | median | 0.323 | 13.598 | 14.301 | 11.764 | **11.541** | 14.811 |
| | mean | 2.792 | 14.071 | 15.032 | **12.560** | 13.402 | 14.743 |
| | std | 6.798 | 6.091 | 5.548 | 6.241 | 6.306 | 3.301 |
| **IRMSD($\downarrow$)** | median | 0.262 | 12.969 | 12.700 | 12.207 | 11.319 | **11.132** |
| | mean | 2.677 | 12.548 | 12.712 | 12.401 | 11.550 | **11.546** |
| | std | 6.803 | 5.435 | 5.390 | 6.353 | 4.681 | 2.258 |
| **DockQ($\uparrow$)** | median | 0.982 | 0.050 | 0.034 | 0.045 | 0.054 | **0.179** |
| | mean | 0.861 | 0.104 | 0.055 | 0.076 | 0.082 | **0.176** |
| | std | 0.310 | 0.172 | 0.067 | 0.090 | 0.084 | 0.064 |
| **Inference time** | | 37328.8 | 197503.1 | 274.5 | 8308.7 | 91.2 | **15.8** |

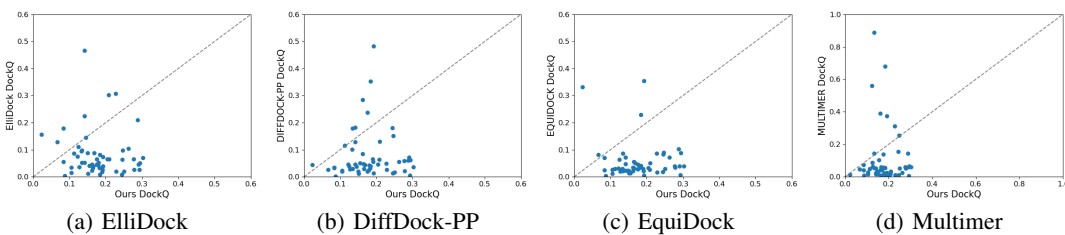

(a) ElliDock      (b) DiffDock-PP      (c) EquiDock      (d) Multimer

Figure 4: Comparative performance of DockQ on SAbDab test set.

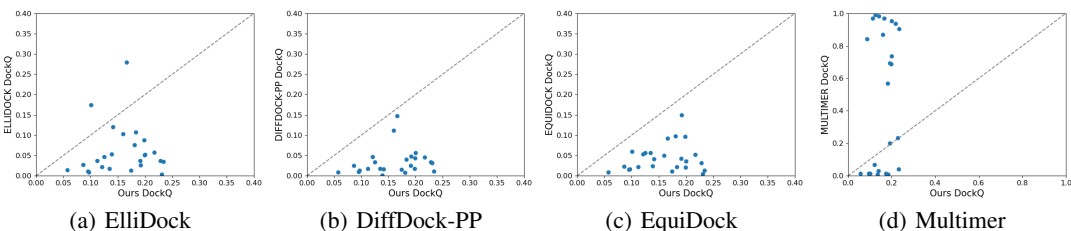

(a) ElliDock      (b) DiffDock-PP      (c) EquiDock      (d) Multimer

Figure 5: Comparative performance of DockQ on DB5.5 test set.

The evaluation of inference time in protein-protein docking models holds significant importance in real-world applications. A efficient inference time enables researchers and practitioners to rapidly screen vast libraries of potential protein-protein interactions. Here we compare the performance of different protein-protein docking methods in terms of inference time on two test sets and results are shown in Tables 1 and 2, accordingly.

As shown in Tables 1 and 2, several key observations emerge: ❶ the traditional search-based docking method HDock exhibits an exceedingly lengthy runtime, owing to the intricate template search process and high computational demands. ❷ Despite being a deep learning model, Multimer still requires additional time for database search to identify similar sequences based on input protein sequences for constructing multiple sequence alignments. Therefore, Multimer is also significantly slower than learning-based methods. ❸ Baseline learning-based models are $10 \sim 1,000$ times faster than HDock and Multimer. Notably, DiffDock-PP is relatively slower among these learning-based method due to the requirement of numerous diffusion steps. ❹ RSDM achieves a notable improvement in inference time in compare with all the baseline methods. The reason for this is that RSDM

simplifies the complexity of model training and enhances the receptor guidance during generation, enabling RSDM to achieve competitive performance with single digit diffusion steps.

## 5.4 ABLATION STUDIES

**Q: Whether the** *personalized sampling distribution* **and the** *step-by-step data purification* **are effective strategies for enhancing the performance of the improved diffusion process?** In this subsection, we carry out an ablation study to analyze the effect of each refinement of RSDM. We consider two variants of RSDM and use DockQ for performance evaluation. The comparison results are shown in Figure 6. From the results, we find that ❶ Considering that higher DockQ scores indicate better performance. The slower convergence of the curve implies superior docking performance. It's evident that RSDM yields the best experimental results. ❷ Between 0 and 0.1, RSDM shows a

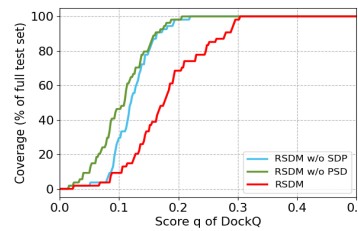

Figure 6: Coverage (% of full test set) of complexes with a Dockq score < q on the SAbDab dataset.

slower slope compared to *RSDM w/o SDP* and *RSDM w/o PSD*, indicating a slower rate of change in coverage for smaller fractions of the test set. This suggests that RSDM achieves poor docking performance less easily, emphasizing the importance of individual refinements in the RSDM. ❸ While *RSDM w/o SDP* and *RSDM w/o PSD* converge similarly at DockQ fractions of 0.15-0.2, *RSDM w/o PSD* has a significantly steeper slope between 0.0-0.1, suggesting that personalized sampling distribution effectively guides ligand prediction by tailoring the noise to maintain receptor specificity. These observations collectively demonstrate the specific contributions of each refinement of RSDM.

## 5.5 HYPER-PARAMETER ANALYSIS

**Q: How is the sensitivity of RSDM to the number of binding sets $Z$?** We evaluate the sensitivity of RSDM to the number of binding sets $Z \in \{20, 40, 60, 80, 100\}$ for training 100 epochs. Figure 7 shows the performance of RSDM with different value of $Z$ on the SAbDab dataset. The results indicate a clear trend of increasing average DockQ performance with the increasing number of binding sites. This result is likely due to the greater number of binding sites providing more interaction points, which enhances the stability and accuracy of the docking process. More binding sites can lead to a stronger and more precise

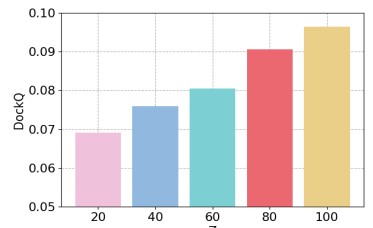

Figure 7: Impact of binding sets quantity $Z$ on average DockQ performance.

binding between the receptor and ligand, thus reflecting in higher DockQ scores.

## 6 CONCLUSION

We develop a novel model for protein-ligand structures generation based on the diffusion model, which is strongly competitive with state-of-the-art learning-based methods. Crucial to the success of the proposed model is to tailor an customized sampling distribution for each ligand and simplify model prediction of raw ligand data through stepwise denoising. RSDM outperforms existing learning-based models and performs competitively against search-based methods at the inference time level. Experimental results on two benchmark datasets and ablation study demonstrate the effectiveness of our proposed model.

In the future, we look forward to explore more sophisticated strategies for incorporating more domain knowledge to refine the reverse process of the protein diffusion model via tailoring customized sampling distributions or investigating additional contextual information. Meantime, the limitation of our model is that RSDM only considers ligand generation without considering the variations in the binding sites, which can affect the ligand generation and binding capabilities. We hope the protein-protein docking paradigm can provide an insight to enhance the flexibility, adaptability, and robustness of our approach to better handle a wider range of receptor-ligand interaction scenarios.

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

## A DERIVATION

Below is a derivation of Eq. 4:

$$
\begin{aligned}
\mathbf{x}_t^{(l)} &= \sqrt{1-\beta_t}\mathbf{x}_{t-1}^{(l)} + \frac{\gamma_t}{T}\bar{\mathbf{x}}^{(r)} + \sqrt{\beta_t}\epsilon_t \\
&= \sqrt{1-\beta_t}\left(\sqrt{1-\beta_{t-1}}\mathbf{x}_{t-2}^{(l)} + \frac{\gamma_{t-1}}{T}\bar{\mathbf{x}}^{(r)} + \sqrt{\beta_{t-1}}\epsilon_{t-1}\right) + \frac{\gamma_t}{T}\bar{\mathbf{x}}^{(r)} + \sqrt{\beta_t}\epsilon_t \\
&= \sqrt{(1-\beta_t)(1-\beta_{t-1})}\mathbf{x}_{t-2}^{(l)} + \sqrt{1-\beta_t}\frac{\gamma_{t-1}}{T}\bar{\mathbf{x}}^{(r)} + \sqrt{1-\beta_t}\sqrt{\beta_{t-1}}\epsilon_{t-1} + \frac{\gamma_t}{T}\bar{\mathbf{x}}^{(r)} + \sqrt{\beta_t}\epsilon_t \\
&= \sqrt{(1-\beta_t)(1-\beta_{t-1})}\left(\sqrt{1-\beta_{t-2}}\mathbf{x}_{t-3}^{(l)} + \frac{\gamma_{t-2}}{T}\bar{\mathbf{x}}^{(r)} + \sqrt{\beta_{t-2}}\epsilon_{t-2}\right) \\
&\quad + \sqrt{1-\beta_t}\frac{\gamma_{t-1}}{T}\bar{\mathbf{x}}^{(r)} + \sqrt{1-\beta_t}\sqrt{\beta_{t-1}}\epsilon_{t-1} + \frac{\gamma_t}{T}\bar{\mathbf{x}}^{(r)} + \sqrt{\beta_t}\epsilon_t \\
&= \sqrt{(1-\beta_t)(1-\beta_{t-1})(1-\beta_{t-2})}\mathbf{x}_{t-3}^{(l)} \\
&\quad + \sqrt{(1-\beta_t)(1-\beta_{t-1})}\frac{\gamma_{t-2}}{T}\bar{\mathbf{x}}^{(r)} + \sqrt{(1-\beta_t)(1-\beta_{t-1})}\sqrt{\beta_{t-2}}\epsilon_{t-2} \\
&\quad + \sqrt{1-\beta_t}\frac{\gamma_{t-1}}{T}\bar{\mathbf{x}}^{(r)} + \sqrt{1-\beta_t}\sqrt{\beta_{t-1}}\epsilon_{t-1} + \frac{\gamma_t}{T}\bar{\mathbf{x}}^{(r)} + \sqrt{\beta_t}\epsilon_t \\
&= \sqrt{\bar{\alpha}_t}\mathbf{x}_0^{(l)} + \sqrt{1-\bar{\alpha}_t}\epsilon + \sum_{i=1}^{t}(\prod_{j=i+1}^{t}\sqrt{1-\beta_j})\frac{\gamma_i}{T}\bar{\mathbf{x}}^{(r)} \\
&= \sqrt{\bar{\alpha}_t}\mathbf{x}_0^{(l)} + \sqrt{1-\bar{\alpha}_t}\epsilon + \sum_{i=1}^{t}(\prod_{j=i+1}^{t}\sqrt{\alpha_j})\frac{\gamma_i}{T}\bar{\mathbf{x}}^{(r)}
\end{aligned}
$$

(12)

## B ALGORITHM

---
**Algorithm 1** Training
---
1: // Forward diffuse
2: $\mathbf{x}_{(1:T)}^{(l)} \sim q(\mathbf{x}_{(1:T)}^{(l)}|\mathbf{x}_0^{(l)}, \bar{\mathbf{x}}^{(r)}) := \mathcal{N}(\mathbf{x}_t^{(l)}; \sqrt{1-\beta_t}\mathbf{x}_{t-1}^{(l)} + \frac{\gamma_t}{T}\bar{\mathbf{x}}^{(r)}, \beta_t\mathbf{I})$
3:
4: // Reverse diffuse
5: $\mathbf{x}_T^{(l)} \sim \mathcal{N}(\bar{\mathbf{x}}^{(r)}, \mathbf{I})$
6: **for** $t = T, \cdots, 1$ **do**
7:     $\mathbf{x}_{t-1}^{(l)} \sim p_\theta(\mathbf{x}_{t-1}^{(l)}|\mathbf{x}_t^{(l)}, \bar{\mathbf{x}}^{(r)}) := \mathcal{N}(\mathbf{x}_{t-1}; \mu_\theta(\mathbf{x}_t^{(l)}, t) - \frac{\gamma_t}{T}\bar{\mathbf{x}}^{(r)}, \Sigma_\theta(\mathbf{x}_t^{(l)}, t)),$
8:     Take gradient descent step on
9:         $\frac{1}{n}\sum_{i=1}^{n}(\mathbf{v}_i - \widetilde{\mathbf{v}}_i)^2$         ▷ Compute the MSE loss
10:         $\frac{1}{2}(y-\widetilde{y})^2$ if $|y-\widetilde{y}| \le \delta$, else $\delta(|y-\widetilde{y}| - \frac{1}{2}\delta)$     ▷ Compute the Huber loss
11:         $\mathbb{E}_q\left[\sum_{t=1}^{T}D_{\text{KL}}(q(\mathbf{x}_{t-1}^{(l)}|\mathbf{x}_t^{(l)}, \mathbf{x}_0^{(l)}, \bar{\mathbf{x}}^{(r)})||p_\theta(\mathbf{x}_{t-1}^{(l)}|\mathbf{x}_t^{(l)}, \bar{\mathbf{x}}^{(r)}))\right]$ ▷ Compute the KL divergence
12: **end for**
13: **return** $\mathbf{x}_0$
---

## C EVALUATION METRICS

We evaluate all models via Complex Root Mean Squared Deviation (CRMSD), Interface Root Mean Squared Deviation (IRMSD) and DockQ (Basu & Wallner, 2016). Specifically, when given both the ground-truth and predicted complex structures, CRMSD is calculated by aligning them with the Kabsch algorithm (Kabsch, 1976) and subsequently computing the CRMSD. Similarly, IRMSD is determined by aligning their interface residues and calculating the RMSD over the interface. DockQ serves as a common metric for protein-protein docking models, represented as a weighted average of three components: contact accuracy, interface RMSD, and ligand RMSD.

# D VISUALIZATION OF CDR-H3

CDR (Complementarity Determining Region) refers to specific regions within antibodies located in the variable regions, primarily responsible for antigen binding. The CDR comprises six variable regions: CDR-H1, CDR-H2, CDR-H3, CDR-L1, CDR-L2, and CDR-L3, where "H" stands for heavy chain and "L" for light chain. While all CDR regions contribute to antigen binding, CDR-H3 is often considered the most critical. This is because CDR-H3 exhibits the highest variability and accounts for much of the specificity, while other CDRs are relatively conserved. In this subsection, we demonstrate the effectiveness of our model by predicting the CDR-H3 region, further highlighting its significance in predicting antibody structures. HERN (Jin et al., 2022) is a recent generative model designed for antibody structure prediction on the CDR-H3 region. We compare our model with HERN and present the comparative performance in Figure 8.

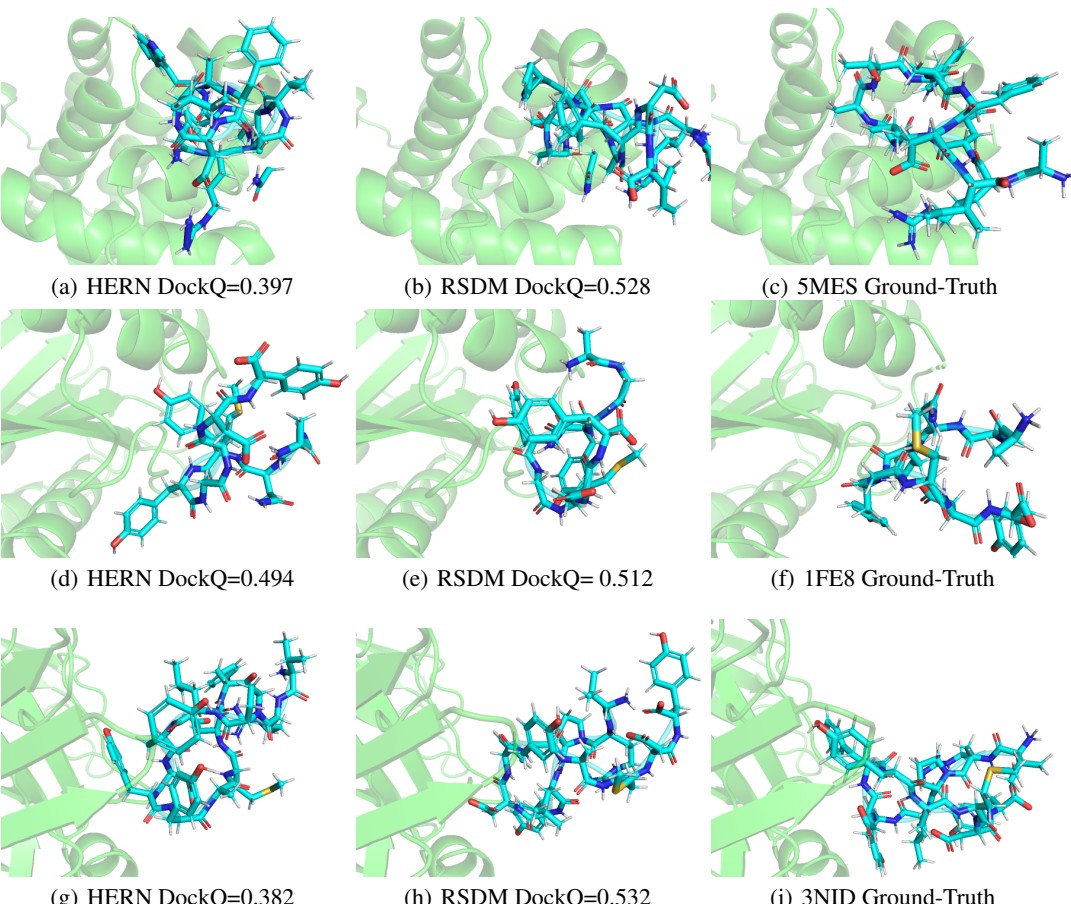

(a) HERN DockQ=0.397          (b) RSDM DockQ=0.528          (c) 5MES Ground-Truth

(d) HERN DockQ=0.494          (e) RSDM DockQ= 0.512          (f) 1FE8 Ground-Truth

(g) HERN DockQ=0.382          (h) RSDM DockQ=0.532          (i) 3NID Ground-Truth

Figure 8: Comparison of visualization results between the structures predicted by HERN and RSDM.

