# OpenReview forum: "Receptor-Specific Diffusion Model: Towards Generating Protein-Protein Structures with Customized Perturbing and Sampling"
_ICLR.cc/2025/Conference — Submitted to ICLR 2025_

### Official Review · Reviewer_xmmV · 2024-11-01

**Soundness:** 3
**Presentation:** 3
**Contribution:** 3
**Rating:** 5
**Confidence:** 4

**Summary:**

The paper presents RSDM, designed to generate protein-ligand structures with receptor-specific properties. Traditional diffusion models often apply a uniform noise distribution during sampling, which fails to capture the unique structural and biochemical distinctions of specific receptors. RSDM addresses this by introducing a customized sampling process that leverages receptor-specific noise distributions, thus creating a more tailored perturbation for each ligand-receptor pair.

**Strengths:**

- RSDM’s introduction of receptor-specific perturbation and sampling in diffusion models is novel, as it directly incorporates receptor-guided noise, enhancing ligand-receptor specificity.
- The model is well-validated on established benchmarks, DB5.5 and SAbDab, and is compared against multiple competitive baselines. The thorough experimental setup, including metrics such as CRMSD, IRMSD, and DockQ, supports the robustness of RSDM and clearly illustrates its advantages in both accuracy and efficiency.
- The paper is clearly structured, with each component of the model explained in detail.
- RSDM’s ability to reduce computational demands while maintaining accuracy could greatly benefit high-throughput docking scenarios, enabling faster and more customized ligand generation.

**Weaknesses:**

- The paper would benefit from a clearer explanation of why an Equivariant Graph Neural Network (EGNN) is particularly well-suited for addressing the protein-protein docking problem. Providing more background on EGNN’s advantages in capturing molecular interactions could better highlight its relevance and strengthen the motivation for its inclusion in this context.
- The process of identifying binding sites and integrating this information into the diffusion model is not fully detailed. It would be more helpful to provide a more comprehensive description of the methods used to determine binding sites and their role in the generation process.

**Questions:**

- In Section 5.4 (Ablation Studies), the authors evaluate model variations primarily through the DockQ metric. It would be better to access the model's performance if the ablation analysis can be extended to include additional performance metrics, such as CRMSD and IRMSD.

---

### Official Review · Reviewer_qs8A · 2024-11-01

**Soundness:** 2
**Presentation:** 3
**Contribution:** 2
**Rating:** 3
**Confidence:** 3

**Summary:**

This paper introduces a receptor-specific diffusion model tailored for the protein-protein docking task. The authors inject the mean position of the receptor pocket into the ligand's prior distribution to improve RMSD for sampled ligand positions. Additionally, the approach includes a network that directly predicts $x_t$ in the diffusion sampling process for this application. Two distinct loss functions are employed for coordinates and structure, enhancing the diffusion model's parameterization. Finally, the model's performance is evaluated against other protein-protein docking baselines using CRMSD, IRMSD, and DockQ metrics.

**Strengths:**

Thanks for the work. The paper is well written with nice figures.

**Weaknesses:**

The key aspects of the diffusion model method are questionable and confusing:

1, For the “receptor-specific” section:
The paper achieves this "receptor" specification by adding the mean position of the binding pocket to the mean of the Gaussian in the prior distribution and gradually decreasing its magnitude during sampling. To me, this approach seems like it is steering the center of the sampled ligand protein to remain in the pocket center. However, in most molecular protein docking and protein-protein docking problems, this can be done by simply removing the Center of Mass (CoM) for the ligand during sampling, which makes it confusing why it is necessary to complicate the process by adding this mean into the prior distribution and modifying the training and sampling processes. Also, Figure 2 shows that incorporating a personalized mean into the sampling distribution reduces the RMSD of alpha carbon considerably. However, I did not see any CoM removal in the training algorithm, so the difference between RSDM and w/o PSD groups in RMSD could simply be because the mean of the ligand is not re-centered.

2, For “step-by-step data purification”:
In Equation 8, $\mathcal{N}(x_{t-1}|\mu_{\theta}(x_t^{(l)},t)-\frac{\gamma_t}{T}x^{(r)},\Sigma_{\theta}(x_t^{(l)},t))$ indicates that $x_{t-1}$ is sampled from this parameterized normal distribution. However, prior to this equation, the authors mention that $x_{t-1}$ is directly predicted by $f_{\theta}(x_t^{(l)},t)$. These two expressions on how to obtain $x_{t-1}$ are contradictory; if you directly predict $x_{t-1}$ from the previous time step, how could it be equivalent to sampling from a parameterized distribution as described in Equation 8? There is no sampling involved if you are directly predicting. I would appreciate if the authors could clarify whether they used sampling to obtain $x_{t-1}$ or if it is deterministically obtained by network prediction.

Furthermore, the authors introduce this “step-by-step data purification” by starting with the criticism, “Such the reverse process (predict $x_0$ with the score network) poses a challenge to the model’s predictive ability and complicates the training process.” I don’t see why this challenges the model’s predictive ability. In diffusion or score-based models, predicting $x_0$, $\epsilon_t$, or $v_t = \alpha_t x_0 + \sigma_t \epsilon$ are three commonly used spaces to parameterize diffusion models. If you directly predict $x_{t-1}$ from $x_{t}$, then your network is no longer directly or indirectly parameterizing the score $\nabla_{x_t} \log P(x_t)$; it is directly predicting a sample instead of the gradient of the probability distribution, so it is no longer a score-based or diffusion model.

Taking a step back, regardless of the confusion discussed above, even if the paper claims that directly predicting $x_t$ is better than $x_0$, I did not see any ablation study showing that predicting $x_t$ improves performance over predicting $x_0$, given the authors' criticism that predicting $x_0$ challenges the model’s predictive ability.

**Questions:**

I would like to learn from the author's perspective from the weaknesses discussed above, thanks!

---

### Official Review · Reviewer_c9wE · 2024-11-03

**Soundness:** 2
**Presentation:** 3
**Contribution:** 2
**Rating:** 3
**Confidence:** 4

**Summary:**

This paper focuses on receptor-ligand binding structure design, introducing personalized sampling distribution and step-by-step data purification in diffusion model to incorporate receptor-specific information. The author address the limitations of previous methods that overlooking the structural and chemical differences between receptors and applying a uniform noise distribution across all receptor types. Specifically, the mean of prior distribution in diffusion process is shited to the mean of the corresponding receptor, and the influence of this information is diminished during the sampling process with a predefined schedule. The author demonstrates that RSDM achieves strong performance among methods without searching, and competitive inference time compared to search-based methods.

**Strengths:**

- RSDM achieves strong performance and inference efficiency compared to methods without searching
- The presentation is mostly clear and the method is easy to follow.

**Weaknesses:**

- The idea of shifting the mean in the personalized sampling distribution has been explored in prior works, such as DecompDiff [1], potentially making the technical contribution a bit weak.
- Missing important baselines, e.g., RFdiffusion [2].

[1] Guan, Jiaqi, et al. "DecompDiff: diffusion models with decomposed priors for structure-based drug design." arXiv preprint arXiv:2403.07902 (2024).

[2] Watson, Joseph L., et al. "De novo design of protein structure and function with RFdiffusion." Nature 620.7976 (2023): 1089-1100.

**Questions:**

1. To clarify the utility of Personalized Sampling Distribution and Step-by-step Data Purification, it would be helpful if the authors could provide more ablation studies detailing the effect on CRMSD, IRMSD, and computational efficiency.
2. The description of the model design is ambiguous and unclear. For example, how does the model handle residue padding when modeling 14 atoms per residue as defined in Section 4.1? More details on implementation would improve reproducibility and readers' understanding of the model design.
3. The baselines are all docking methods. A comparison with generative models, such as RFdiffusion, would provide a more comprehensive evaluation.

---

### Official Review · Reviewer_vmR6 · 2024-11-03

**Soundness:** 2
**Presentation:** 1
**Contribution:** 2
**Rating:** 3
**Confidence:** 3

**Summary:**

This paper addressed challenges in protein-ligand structure design facilitated by generative models, specifically targeting inefficiencies and generalizations in existing diffusion-based approaches. The authors proposed the Receptor-Specific Diffusion Model (RSDM), which introduces a novel method of customized perturbing and sampling to more accurately generate ligands tailored to specific receptors. RSDM uses receptor-specific information to adjust the sampling distribution, altering noise for customized perturbations, and employs a stepwise denoising schedule to refine ligand generation. Experimental results demonstrate that RSDM is highly competitive with leading models like ElliDock and DiffDock-PP, while also offering faster inference speeds. This positions RSDM as a promising tool for reliable and efficient protein-ligand generation.

**Strengths:**

The motivation behind this work is clear and robust. Receptor-specific diffusion processes incorporate more informative prior knowledge for modeling binding structures, with the center of binding site atoms serving as an effective indicator for binding.

The proposed method outperforms all baseline models across various metrics on the DB5.5 dataset. Additionally, it boasts impressive inference speed.

**Weaknesses:**

The presentation of this work could be improved as it contains some typographical errors; for instance, 'sets' on line 513 may be a typo.

The method itself is relatively straightforward, with similar approaches employed in DecompDiff [1].
Informative priors are used to refine the diffusion and reverse processes, enhancing the quality of generated samples, although some important references are missing.

Furthermore, since the binding site is unknown during inference, it would be beneficial to investigate how the quality of the predicted binding site affects performance.

The proposed method underperforms on antibody structure prediction compared to several baselines in terms of IRMSD.

The generalizability of this approach could be further validated by extending the framework to protein-ligand (small molecule) complex structure prediction tasks.

Reference:

[1] Guan, J., Zhou, X., Yang, Y., Bao, Y., Peng, J., Ma, J., Liu, Q., Wang, L. and Gu, Q., 2024. DecompDiff: diffusion models with decomposed priors for structure-based drug design. ICML 2023.

**Questions:**

See the weaknesses above.

**Details Of Ethics Concerns:**

No ethics concerns.

---

### Meta-Review · Area_Chair_f1vc · 2024-12-20

**Metareview:**

The reviewers all recommended to reject this paper, and the authors have not submitted a rebuttal. Therefore I recommend to reject this paper.

**Additional Comments On Reviewer Discussion:**

There was no reviewer discussion given the absence of a rebuttal.

---

### Decision · Program_Chairs · 2025-01-22

Reject